# ROBUST SPARSE AND DENSE MATCHING

## ABSTRACT

Finding corresponding pixels within a pair of images is a fundamental computer vision task with various applications. Due to the specific requirements of different tasks like optical flow estimation and local feature matching, previous works are primarily categorized into dense matching and sparse feature matching focusing on specialized architectures along with task-specific datasets, which may somewhat hinder the generalization performance of specialized models. In this paper, we propose RSDM, a robust network for sparse and dense matching. A cascaded GRU module is elaborately designed for refinement to explore the geometric similarity iteratively at multiple scales following an independent uncertainty estimation module for sparsification. To narrow the gap between synthetic samples and real-world scenarios, we organize a new dataset with sparse correspondence ground truth by generating optical flow supervision with greater intervals. In due course, we are able to mix up various dense and sparse matching datasets significantly improving the training diversity. The generalization capacity of our proposed RSDM is greatly enhanced by learning the matching and uncertainty estimation in a two-stage manner on the mixed data. Superior performance is achieved for zero-shot matching as well as downstream geometry estimation across multiple datasets, outperforming the previous methods by a large margin.

## 1 INTRODUCTION

Correspondence matching is a fundamental task in computer vision with various applications like Simultaneous Localization and Mapping (SLAM), geometry estimation, and image editing. Due to the specific requirements of different applications, the recent learning-based matching works are commonly categorized into two branches: sparse and dense matching. The primary difference is that dense matching, like optical flow estimation, stereo matching, and multi-view stereo matching, is required to provide the matching estimation for each pixel even in occluded regions, while sparse matching is only responsible for finding the corresponding key points given a pair of images. Besides, in the context of dense matching, the image pairs to be matched typically have limited changes of viewpoint with a relatively small temporal interval while the image pairs for sparse matching normally have more significant changes in viewpoint with various image properties.

Oriented by different applications and requirements, the research of dense and sparse matching has followed separate paths for a long time. Benefiting from the accumulating customized datasets, the learning-based approaches witness remarkable improvement for particular tasks like optical flow estimation(Teed & Deng, 2020; Huang et al., 2022; Sun et al., 2017; Xu et al., 2022b), stereo matching (Zhang et al., 2021; Xu & Zhang, 2020; Chang & Chen, 2018; Xu et al., 2022a), and geometry estimation(Sun et al., 2021; Chen et al., 2022; Junjie Ni, 2023; Edstedt et al., 2023). However, as discussed in (Truong et al., 2020), the generalization capacity of specialized matching models may be limited when new scenarios especially those with large displacements are applied. Some pioneering works (Truong et al., 2020; 2021; Shen et al., 2020; Xu et al., 2023; Li et al., 2020; Melekhov et al., 2019) have already made attempts to improve the robustness of matching problems by proposing a universal matching framework capable of both sparse and dense matching.

Despite the delightful improvement in the generalization performance, the proposed universal matching networks still struggle in the performance of specific tasks compared with the specialized models. Besides, how to exploit the advantages of multiple task-specific datasets to boost the robustness of matching is still under exploration. It has been revealed that increasing the diversity of training data brings a tremendous improvement in generalization capacity for monocular depth

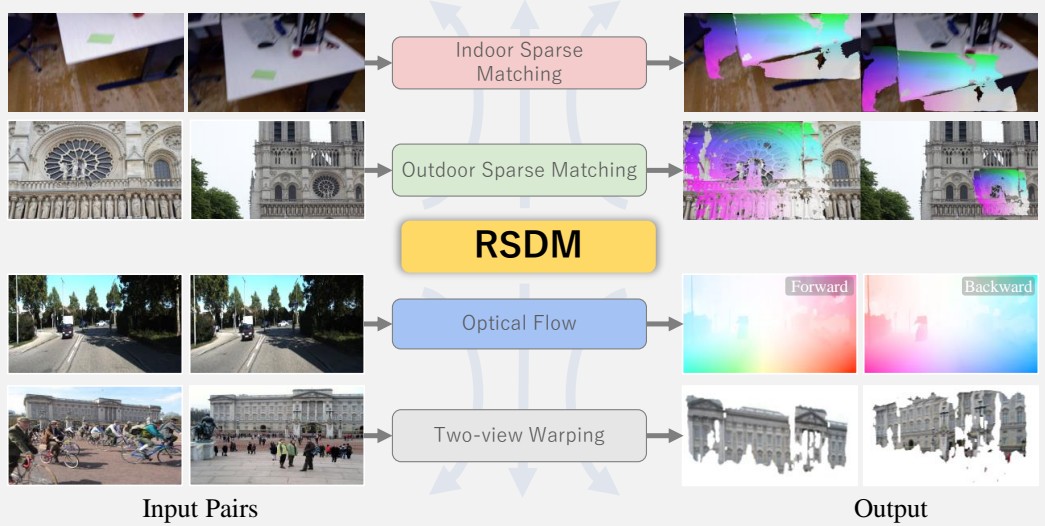

Figure 1: **An illustration of the general capacity of our RSDM.** RSDM is a robust matching model capable of indoor and outdoor sparse matching even in cases of great displacement or blurred input images. The matched features are indicated by the same color. Moreover, we can also handle the all-paired dense matching like optical flow estimation and the two-view reconstruction by warping the corresponding pixels to the target view.

estimation(Bhat et al., 2023; Yin et al., 2020; Wei Yin, 2023; Ranftl et al., 2022; Truong et al., 2021). However, the datasets used for dense and sparse correspondence matching have obvious domain gaps. Constrained by the difficulty in obtaining the fine-grained annotations per pixel for dense correspondence matching, most specialized models follow the training pipeline of first pre-training on a large-scale synthesized dataset while fine-tuning on a small-scale real-world dataset. Sparse feature matching tasks for geometry estimation, on the contrary, have an extensive amount of real-world training data (Li & Snavely, 2018; Dai et al., 2017) but with relatively coarse pixel-wise annotations. Directly training on a mixture of multiple task-specific datasets varying in the domain may hamper the matching performance and impede the downstream geometry estimation.

To tackle the above-mentioned challenges, we propose a **r**obust **s**parse and **d**ense **m**atching network termed RSDM. To be specific, we construct a dense matching framework with a cascaded GRU refinement to fully exploit the geometry similarly with fine-grained features across multiple scales. An independent uncertainty estimation module is also constructed for sparsification. We mix up multiple optical flow and sparse matching datasets for numerous data to train our model in a decoupled manner so that the perturbation introduced by diverse domains can be alleviated. We additionally organize a dataset with sparse correspondence ground truth based on the TartanAir (Wang et al., 2020) dataset to obtain a closer distribution of displacement to the real world. It is worth mentioning that we build the cascaded GRU refinement to take advantage of the fine-grained features at a higher resolution. We believe the state-of-the-art methods for optical flow estimation (Teed & Deng, 2020; Huang et al., 2022; Shi et al., 2023; Xu et al., 2022b) could fit in our framework as well. The contributions can be summarized as follows:

- We propose a robust sparse and dense matching network termed RSDM which can generalize well to unseen scenarios with our proposed cascaded GRU refinement for dense correspondence estimation and an uncertainty estimation module for sparsification.

- We explore the effectiveness of scaling up the training data by mixing up multiple datasets. A comprehensive analysis is conducted to explore a more reasonable training strategy for enhanced robustness.

- Our RSDM achieves state-of-the-art generalization performance in zero-shot evaluations for both matching and geometry estimation across multiple datasets, outperforming previous generalist and specialized models by an obvious margin.

## 2 RELATED WORKS

### 2.1 SPARSE FEATURE MATCHING

For a long time, the geometry estimation problem has been dominated by sparse correspondence matching methods. The classic methods(Rublee et al., 2011; Liu et al., 2010) propose robust hand-crafted local features for matching and have been adopted in many 3D reconstruction related tasks. Following the manner of detection and match, the leaning-based methods efficiently improve the matching accuracy, among which SuperGlue (Sarlin et al., 2020) is a representative network. Given two sets of interest points as well as the corresponding descriptors, SuperGlue utilizes a transformer-based graph neural network for feature enhancement and to obtain great improvement. LightGlue (Lindenberger et al., 2023) further modifies the SuperGlue by proposing an adaptive strategy due to the matching difficulty which effectively accelerates the inference. Since the proposal of LoFTR (Sun et al., 2021), the detector-free local feature matching method which discards the feature detector stage appeals to great attention(Wang et al., 2022; Chen et al., 2022; Tang et al., 2022). LoFTR (Sun et al., 2021) takes a coarse-to-fine strategy by fist establishing a dense matching correspondence and removing the unreliable matches at the refinement stage. Self-attention and cross-attention with the transformer are introduced to enlarge the receptive field. Several works are improved upon LoFTR like ASpanFormer(Chen et al., 2022) which adopts a novel self-adaptive attention mechanism guided by the estimated flow while QuadTree (Tang et al., 2022) primarily focuses on the optimization of attention mechanism by selecting the sparse patches with the highest top $K$ attention scores for attention computation at the next level so that the computation cost can be efficiently reduced. Sparse correspondence matching plays an important role in geometry estimation like pose estimation, 3D reconstruction, but the sparsification of matching estimation impedes its applications when all-paired matches are required. Recently, some sparse matching works(Edstedt et al., 2023; Junjie Ni, 2023; Li et al., 2020; Truong et al., 2021; 2023) are constructed on the base of dense matching where the all-paired matching results are preserved along with a selecting module for sparsification. This is a great step towards the generalist matching model.

### 2.2 DENSE MATCHING

In the context of dense correspondence matching, it is normally categorized into multiple specific tasks containing stereo matching (Zhang et al., 2021; Chang & Chen, 2018; Xu & Zhang, 2020; Li et al., 2022; Xu et al., 2022a; Lipson et al., 2021), multi-view stereo matching(Ma et al., 2022b), and optical flow(Dosovitskiy et al., 2015; Ilg et al., 2017; Sun et al., 2017; Teed & Deng, 2020; Xu et al., 2022b; Huang et al., 2022; Shi et al., 2023; Sui et al., 2022), *etc.* Dense correspondence matching is required to provide the matching prediction per pixel even in occluded regions which are discarded in the sparse matching problem. Among all the dense matching, the optical flow estimation is relatively more comprehensive due to the disordered motions. The pioneering work RAFT(Teed & Deng, 2020) proposes a GRU-based iterative mechanism for refining the estimated optical flow by looking up the correlation pyramid repeatedly. The proposal of RAFT has been modified to various dense matching tasks besides the optical flow estimation(Huang et al., 2022; Shi et al., 2023; Jiang et al., 2021; Dong et al., 2023) including mvs(Ma et al., 2022a), stereo matching(Li et al., 2022; Lipson et al., 2021), which validates its capacity as a universal architecture for dense matching. The limitation for dense matching lies in that only limited real-world datasets with constrained variation in perspectives are available which may hamper the generalization performance to some extent.

### 2.3 GENERALIST CORRESPONDENCE MATCHING

To mitigate the limitation of specialized matching models and unify the matching problems, some attempts have been proposed. MatchFlow(Dong et al., 2023) manages to improve the robustness of optical flow estimation by utilizing a model pretrained on real-world datasets. UniMatch (Xu et al., 2023) proposes to unify the estimation for optical flow, stereo matching, and depth estimation. However, they still can't handle sparse feature matching tasks or significant changes of viewpoint in the real world. DKM (Edstedt et al., 2023) and PATS (Junjie Ni, 2023) are competent for unified matching but they utilize solely the sparse matching datasets. PDCNet and PDCNet+ (Truong et al., 2021; 2023) propose a universal matching framework with training on both sparse matching datasets and their synthesized optical flow datasets. The limitation of PDCNet+ lies in that the synthesized optical flow couldn't simulate the real changes in perspective and motions in real-world scenarios.

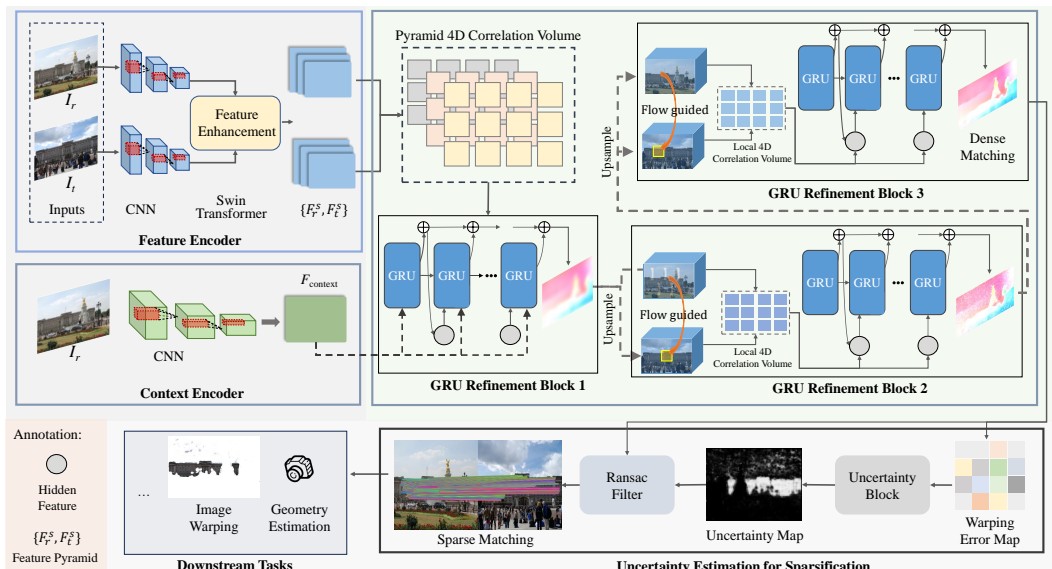

Figure 2: An overview of our proposed RSDM which consists of a pure dense matching network with our elaborate cascaded GRU refinement(the upper-right part in shallow green) along with an independent uncertainty estimation module for sparsification. Some detailed operations like the skip connection are omitted here for visual clarity.

Moreover, the influence of joint learning of both matching and uncertainty on the mixed datasets of diverse domains remains unexplored.

## 3 METHOD

### 3.1 MATCHING NETWORK

As discussed above, the target distribution of displacement magnitude differs in various matching tasks. To unify both dense matching and sparse feature matching, it's crucial to exploit global and local geometry correspondence which requires elaborate design. We formulate the universal matching as a dense matching problem following an uncertainty estimation module for sparsification. The dense matching $f$ is defined in the definition of optical flow.

The milestone work of RAFT (Teed & Deng, 2020) provides an appealing approach as the multiscale correlation pyramid effectively enlarges the receptive field and the iterative looking-up operation associated with contextual information helps with the local correlation exploration. The vanilla RAFT as well as the following modifications (Huang et al., 2022; Shi et al., 2023), however, suffer from the utilization of a coarse feature at 1/8 resolution which leads to an inevitable loss of fine-grained features. To alleviate this, we first construct a feature pyramid network to obtain a set of reference and target features $\{F_r^s, F_t^s\}$ at the corresponding scales $s$ of $\{1/8, 1/4, 1/2\}$ as illustrated in the upper-right block of Fig3. We adopt self-attention and cross-attention based on the swin-transformer for feature enhancement as GMFlow (Xu et al., 2022b) at the first two scales.

To adapt to higher resolutions, a cascaded GRU refinement module is proposed. Instead of building all-paired correlations across every scale (Li et al., 2022; Jahedi et al., 2022), the all-paired correlation volume $C_{full}$ is only introduced at the coarsest resolution of 1/8 while the subsequent correlation volume is formulated locally. The all-paired correlation volume $C_{full}$ is formulated via the dot-product operation as follows:

$$C_{full} = \frac{F_r \cdot F_t^T}{\sqrt{D}} \in \mathbb{R}^{\frac{H}{8} \times \frac{W}{8} \times \frac{H}{8} \times \frac{W}{8}},\tag{1}$$

where $(H, W)$ is the spatial resolution of the original image and D denotes the feature dimension. A correlation pyramid is then constructed with additional average pooling operation as RAFT. Given

the current flow estimation $f(x)$ at grid $x$, the local correlation with the radius $r$ is built as:

$$C_{local}(x) = \frac{F_r(x) \cdot F_t^T(\mathcal{N}(x + f(x))_r)}{\sqrt{D}},$$

(2)

where $\mathcal{N}(x)_r$ is the set of local grids centered at $x$ within radius $r$ defined as:

$$\mathcal{N}(x)_r = \{x + dx | dx \in \mathbb{Z}^2, ||dx||_1 \leq r\}$$

(3)

Rather than initializing the hidden status at each scale (Jahedi et al., 2022; Li et al., 2022), we upsample hidden features with bilinear interpolation and pass it to the next refinement stage. Given the correlation as well as the contextual information, we compute motion features as the vanilla RAFT and feed it to cascaded GRU refinement for flow residual estimation which is then used for updating the matching flow in an iterative manner. We utilize L1 loss for supervision across multiple scales between the matching prediction and ground truth:

$$L_m = \sum_{s=1}^{S} \gamma_s \begin{cases} ||f_s - f_{gt}||_1 & \text{if dense matching} \\ ||f_s - f_{gt}||_1 \odot p & \text{if sparse matching} \end{cases},$$

(4)

where $p$ indicates the valid mask where sparse correspondence ground truth is available and $\lambda_s$ is a scalar for adjusting the loss weight at scale $s$. We supervise all predicted matches on the optical flow datasets and valid estimated matches sparse feature matching datasets.

## 3.2 DECOUPLED UNCERTAINTY ESTIMATION

The indispensable component that unifies the matching problem is the uncertainty estimation module(Truong et al., 2021; Edstedt et al., 2023; Truong et al., 2023; Li et al., 2020). Conventional methods which estimate the valid mask simultaneously with correspondence yield unsatisfactory performance when trained on multiple dataset jointly. We argue that the joint learning strategy may introduce inevitable noise as the valid mask is closely associated with the matching prediction being inaccurate and ambiguous, especially at the early stage of training. It is an ill-posed problem to determine valid areas given predicted matches of low quality. Moreover, the domain of invalid matches that are filtered out for downstream pose estimation may differ from the occluded regions in optical flow estimation. As we will discuss in the later section4.2, the joint learning of matching and uncertainty could impede the performance of one or both tasks, especially when training on a mixture of datasets.

To alleviate this problem, we decouple our universal matching model into a pure dense correspondence network defined in the subsection3.1 accompanied by an independent uncertainty estimation network. As shown in Fig3, after obtaining the dense correspondence results, the matching network is frozen. We compute the difference by warping the feature map and RGB image of the target view to the reference view according to the estimated flow. The uncertainty prediction $\hat{p}$ is then computed by feeding the warping difference to a shallow convolution network and supervised by valid mask ground truth $p$ with binary cross-entropy loss following previous works:

$$L_u = \sum_{x,y} p \, log(\hat{p}) + (1 - p)log(1 - \hat{p})$$

(5)

The uncertainty module is only applied at the sparsification stage for the downstream geometry estimation task. We follow the balanced sampling strategy proposed in DKM (Edstedt et al., 2023) to sample valid points within the uncertainty threshold for essential metric calculation.

## 3.3 SYNTHESIZED OPTICAL FLOW WITH LARGE INTERVALS

To further exploit the advantages of fine-annotated synthesized datasets but with significant displacement, we randomly sample frames with great intervals from 15 to 30 on the TartanAir dataset. Given the provided intrinsic metrics as well as the depth, we first project the reference frame from 2D pixels to 3D point clouds and then reproject to 2D pixels of the target view according to the extrinsic metric. The matching ground truth $f_{r \to t}$ is computed as the difference between the projected and original pixel coordinates. To access the valid mask $p$ for sparse supervision, the same procedure

is repeated from the target view to the reference view and obtain the inverse matching ground truth $f_{t \to r}$ so that the forward-backward consistency check (Meister et al., 2018; Xu et al., 2022b) can be applied:

$$
\begin{aligned}
p =& |f_{r \to t}(x) + f_{t \to r}(x + f_{r \to t}(x))|^2 \\
& < \alpha_1(|f_{r \to t}(x)|^2 + |f_{t \to r}(x + f_{r \to t}(x))|^2) + \alpha_2,
\end{aligned}
\tag{6}
$$

where $\alpha_1$ is 0.05 and $\alpha_2$ is 0.5. We synthesize around 700K training data pairs over 369 scenarios to construct a new dataset named TartanAir Sampled(TS). Note that the synthesized dataset is only utilized at the stage of matching with sparse ground truth. More details can be found in the appendixA.1.

## 4 EXPERIMENT

### 4.1 IMPLEMENTATION DETAILS

Our proposed RSDM is trained in a two-stage manner. At the stage of correspondence learning, we first utilize the Megadepth(M) (Li & Snavely, 2018) dataset with sparse correspondence ground truth for 200K iterations on a sub-split of 1.4 million pairs of data. Then we collect additional training data containing the ScanNet(Sc) (Dai et al., 2017), FlyingThings3D(T) (Mayer et al., 2016), and original TartanAir(TA) (Wang et al., 2020) datasets as well as our generated TartanAir Sampled dataset (TS) along with the Megadepth, reaching a total amount of around 4 million pairs of training data. The network is trained for 240K iterations on the mixture dataset. We follow GMFlow (Xu et al., 2022b) with the setting of the AdamW (Loshchilov & Hutter, 2017) optimizer and the cosine learning rate schedule as well as the data augmentation for optical flow related datasets(T,TA,TS). For both the Megadepth and ScanNet datasets, the input training images are directly resized to $384 \times 512$ while the rest training data are cropped to the same resolution. The training batch size is set to 16 with a learning rate of 2e-4 for the above two rounds of training. We further finetune the matching network for another 200K iterations at the resolution of $512 \times 704$ with a batch size of 8 and the initial learning rate is decreased to 1e-5. At the stage of uncertainty learning, the parameters of the dense matching network are frozen. We train the uncertainty estimation module on the mixture of both Megadepth and ScanNet datasets for 2 epochs with a batch size of 4 and the learning rate is 1e-4. The training of our RSDM is conducted on the NVIDIA-RTX-4090 GPUs. The related GRU iterations are $\{7, 4, 2\}$ as we gradually recover the resolution for training and ablation experiments with a corresponding searching radius of $\{4, 4, 2\}$. When comparing with other approaches, the iterations increase to $\{12, 12, 2\}$ which is a closer amount of refinement to RAFT-based methods(Huang et al., 2022; Teed & Deng, 2020; Li et al., 2022; Sui et al., 2022).

**Evaluation metrics:** We report the average end-point-epe (AEPE, lower the better) and percentage of correct key points (PCK-T, higher the better) within a specific pixel threshold T. F1 metric(lower the better) is reported for the KITTI (Menze & Geiger, 2015) dataset which depicts the percentage of outliers averaged over all valid pixel of the dataset. For pose estimation, we follow previous works (Truong et al., 2021; Sun et al., 2021; Edstedt et al., 2023; Junjie Ni, 2023) by solving the essential matrix given the corresponding pixels. The accuracy is measured by AUC (higher the better) across different thresholds ($5^{\circ}, 10^{\circ}, 20^{\circ}$).

**Evaluation datasets:** To validate the generalization performance of our RSDM, we perform zero-shot evaluations on multiple benchmark datasets containing the ETH3D (Schöps et al., 2017), HPatches (Balntas et al., 2017), KITTI (Menze & Geiger, 2015), and TUM (Sturm et al., 2012) datasets for correspondence estimation. The downstream pose estimation is conducted on the TUM (Sturm et al., 2012) and YFCC (Thomee et al., 2016) datasets. The geometry estimation results are also reported on the ScanNet (Dai et al., 2017) and Megadepth (Li & Snavely, 2018) datasets. We further adopt the Sintel (Butler et al., 2012) dataset to compare the performance of optical flow estimation under a standard training setting as can be found in the appendixA.3.

### 4.2 ABLATION STUDY

**Network architecture and training strategy:** As presented in Tab4.2, we analyze the effectiveness of the feature enhancement and the training strategy. The baseline is set as the joint learning of both matching and uncertainty with feature enhancement on the Megadepth dataset (Li & Snavely,

Table 1: **Ablation experiments on the training components and strategies**. $W$ indicates the feature enhancement is utilized while $WO$ represents removing it. It is clear that the independent learning strategy improves generalization performance in matching and pose estimation with the help of feature enhancement.

| Method | Dataset | Pose Estimation | | Correspondence Matching | | |
|---|---|---|---|---|---|---|
| | | Megadepth | YFCC | Megadepth | HPatches | ETH3D |
| | | AUC@5$^\circ$ | AUC@5$^\circ$ | PCK-1 | AEPE | AEPE |
| Joint-Learning | | | | | | |
| WO | M | 54.3 | 48.0 | 60.8 | 74.5 | 2.3 |
| W | M | 55.3 | 48.2 | **78.3** | 31.1 | 4.2 |
| W | M + TS | 51.9 | 47.8 | 77.4 | 19.2 | 2.2 |
| Decoupled-Learning | | | | | | |
| W | M | **55.7** | **48.5** | 78.2 | 24.9 | 2.3 |
| W | M + TS | 54.0 | 48.2 | 77.3 | **16.6** | **2.0** |

Table 2: **Ablation study on the effectiveness of scaling up training data.** We perform zero-shot matching evaluations as we gradually scale up the training data for diversity increment. Obviously, the employment of numerous training data brings significant generalization improvement.

| Dataset | HPatches | | KITTI | | ETH3D | |
|---|---|---|---|---|---|---|
| | AEPE | PCK-1 | AEPE | F1 | AEPE | PCK-1 |
| C+T | 55.3 | 38.3 | 5.4 | 13.9 | 3.9 | 55.0 |
| M | 24.9 | 44.8 | 12.6 | 18.4 | 2.2 | 50.6 |
| M+Sc | 15.3 | 42.8 | 10.8 | 17.7 | **2.0** | 54.1 |
| M+Sc+T+TA | **13.1** | 44.3 | 4.1 | 10.8 | **2.0** | 55.9 |
| M+Sc+T+TA+TS | 13.3 | **46.3** | **3.5** | **9.6** | **2.0** | **56.3** |

2018). Obviously, after removing the self-attention and cross-attention, the network suffers a significant degeneration in matching performance which validates its importance. Directly training on the mixed datasets may improve the matching generalization, but the pose estimation suffers an obvious drop on both the Megadepth and YFCC (Thomee et al., 2016) datasets although the matching accuracy remains almost unchanged on the Megadepth, which we attribute to the domain gap between the synthesized images and real-world data for uncertainty estimation. To validate our assumption, we decouple the learning process by learning an independent uncertainty module detached from the learning of similarity. The matching network is trained separately on the Megadeth and mixed datasets, respectively. The uncertainty estimation is next trained solely on the Megadepth dataset. The advantage of the decoupled training strategy is obvious as the matching performance achieves further promotion reaching the lowest AEPE metric(16.6) on the HPatches dataset (Balntas et al., 2017). Compared with the joint learning on the mixed datasets, the decoupled training strategy improves the AUC5$^\circ$ metric from 51.9 and 47.8 to 54.0 and 48.2 on the Megadepth and YFCC datasets, respectively.

**Scaling up training data**. It is clear from Tab4.2 that as we scale up the diversity of training data, the generalization performance of correspondence matching improves significantly. Due to the synthesized training data (Dosovitskiy et al., 2015; Mayer et al., 2016) and limited changes of viewpoint, the poor generalization performance from the optical flow estimation is within expectation. The accuracy improves obviously as the training datasets switch to the real-world Megadepth (Li & Snavely, 2018). We then finetune the model on the mixture of different datasets for 1 epoch separately. The introduction of the real-world indoor ScanNet (Dai et al., 2017) dataset brings overall improvements on the HPatches (Balntas et al., 2017) and ETH3D (Schöps et al., 2017) datasets indicated by the decreasing average end-point error. When the optical flow datasets are further mixed up, the average end-point-error and percentage of outliers accordingly drop from 10.8 to 4.1 and 17.7% to 10.8% on the KITTI dataset (Menze & Geiger, 2015), and the PCK-1 metric climbs to

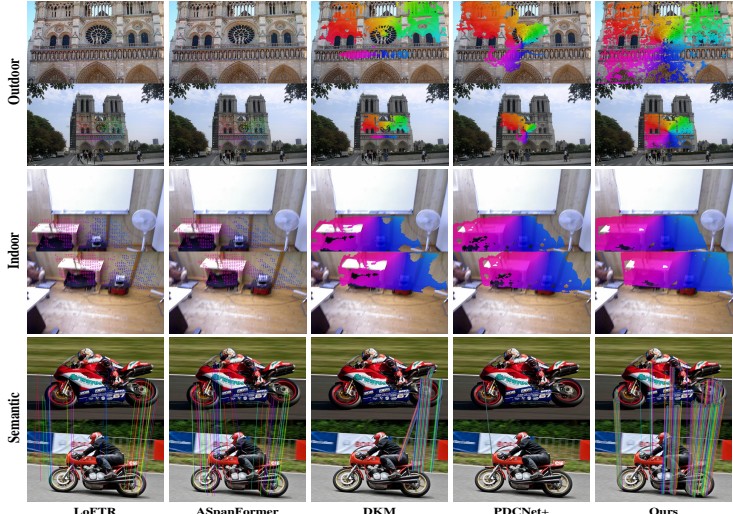

Figure 3: **Visualized comparisons.** Our RSDM shows superior performance by obtaining more matches given a pair of indoor or outdoor images. Moreover, our method shows the potential for robust semantic correspondence as well. The same color indicates the matched features.

55.9% on the ETH3D datasets. The performance on the HPatches dataset obtains further boost as the AEPE metric reaches the lowest result. This improvement of generalization over every testing dataset emphasizes the importance of collecting fine-annotated optical flow datasets to scale up the diversity. After the employment of our synthesized dataset, the PCK-1 metric reaches 46.3% on the HPatches and the percentage of outliers decreases to 9.6% on the KITTI dataset, surpassing the baseline by 47.8%. Our synthesized TartanAir Sampled dataset also helps to obtain the best matching performance on the ETH3D dataset whose PCK-1 metric rises from 50.6% to 56.3% compared with training solely on the Megadepth dataset.

### 4.3 COMPARISONS WITH OTHER SOTAS

We divide the competing matching methods into two categories: specialized models and generalist models depending primarily on their targeted tasks and the training data. Only sparse ground truth in valid regions is available on the HPatches (Balntas et al., 2017) and ETH3D (Schöps et al., 2017) datasets while the occluded regions are also taken into consideration on the KITTI (Menze & Geiger, 2015) dataset.

**Zero-shot Correspondence Matching.** We conduct experiments to compare the generalization capacity for zero-shot evaluation. The specialized models for optical flow estimation achieve relatively better performance on the KITTI dataset due to the all-paired supervision strategy and the densely annotated datasets but suffer a significant degeneration when switching to real-world scenarios as reflected by the great average end-point error and the poor percentage of correct matches. The accuracy on the HPatches dataset gets promoted when the geometry estimation methods trained on the Megadepth dataset are adopted whose AEPE drops from 60.8% to 27.1% for DKM compared with RAFT and increases the PCK-1 by 7%. However, the limitation of geometry estimation works lies in their performance on the KITTI dataset due to the incapacity of occluded regions and the relatively coarse annotations. Taking advantage of training data with greater diversity, the generalist models show a balanced performance. Among all the competing methods, our proposed RSDM reaches the best generalization performance over all three datasets. Compared with the second best method PDCNet+ (Truong et al., 2023), our RSDM improves the PCK-1 from 44.9% to 47.9% and 53.3% to 56.4% on the HPatches and ETH3D dataset, respectively. Our universal matching work still obtains the best performance on the KITTI dataset.

**Geometry Estimation.** We perform pose estimation on the TUM (Sturm et al., 2012) and YFCC (Thomee et al., 2016) datasets for zero-shot evaluation. The performance on the ScanNet (Dai et al., 2017) dataset is also reported in Tab4.3. The evaluation of the TUM dataset is only conducted

Table 3: **Comparison with other methods for zero-shot matching evaluations**. Specialized models for optical flow estimation and dense-based geometry estimation methods as well as the universal matching models are compared on multiple datasets. Our proposed RSDM achieves the best performance among all the competing approaches. Only the underlined results are obtained with optical flow models trained on the FlyingChairs and FlyingThings3D datasets while others are tested with additional training for Sintel submission. * indicates we utilize the officially released model and code for evaluations at the original resolution except that we fix the max resolution to $920 \times 1360$ for FlowFormer (Huang et al., 2022) and $860 \times 1260$ for FlowFormer++ (Shi et al., 2023) due to the computation memory cost. Syn represents the customized synthetic datasets.

| Method | Task | HPatches | | TUM | | ETH3D | | KITTI |
|---|---|---|---|---|---|---|---|---|
| | | AEPE | PCK-1 | AEPE | PCK-1 | AEPE | PCK-1 | F1 |
| specialized models | | | | | | | | |
| RAFT* (Teed & Deng, 2020) | OF | 60.8 | 36.0 | 8.5 | 11.6 | 6.7 | 48.7 | 17.4 |
| FlowFormer* (Huang et al., 2022) | OF | 81.8 | 31.7 | 7.4 | 11.5 | 4.9 | 47.8 | 14.7 |
| FlowFormer++* (Shi et al., 2023) | OF | 81.4 | 28.5 | 6.8 | 11.4 | 3.5 | 48.1 | 14.1 |
| UniMatch* (Xu et al., 2023) | OF | 40.5 | 37.6 | 6.5 | 11.6 | 3.5 | 50.1 | 17.6 |
| DKM* (Edstedt et al., 2023) | GE | 19.0 | 34.7 | 6.1 | 10.3 | 2.2 | 50.1 | 21.0 |
| generalist models | | | | | | | | |
| GLUNet* (Truong et al., 2020) | - | 25.1 | 39.6 | 6.7 | 10.4 | 4.4 | 31.6 | 37.5 |
| PDCNet+(D)* (Truong et al., 2023) | - | 17.5 | 44.9 | 4.9 | 11.5 | 2.3 | 53.3 | 12.6 |
| Ours | - | **8.8** | **47.9** | **4.1** | **12.3** | **2.0** | **56.4** | **10.9** |

Table 4: **Downstream pose estimation**. We conduct geometry estimation on the YFCC and TUM datasets for zero-shot evaluations. Results on the ScanNet validation set are also reported. Our method achieves the best performance for generalization comparisons.

| Method | ScanNet(AUC) | | | YFCC(AUC) | | | TUM(AUC) | | |
|---|---|---|---|---|---|---|---|---|---|
| | @5° | @10° | @20° | @5° | @10° | @20° | @5° | @10° | @20° |
| LoFTR (Sun et al., 2021) | 22.0 | 40.8 | 57.6 | 42.4 | 62.5 | 77.3 | - | - | - |
| DRCNet (Li et al., 2020) | 7.7 | 17.9 | 30.5 | 29.5 | 50.1 | 66.8 | - | - | - |
| MatchFormer (Wang et al., 2022) | 24.3 | 43.9 | 61.4 | 53.3 | 69.7 | 81.8 | - | - | - |
| ASpanFormer (Chen et al., 2022) | 25.6 | 46.0 | 63.3 | 44.5 | 63.8 | 78.4 | - | - | - |
| PATS (Junjie Ni, 2023) | 26.0 | 46.9 | 64.3 | 47.0 | 65.3 | 79.2 | - | - | - |
| DKM (Edstedt et al., 2023) | **29.4** | **50.7** | **68.3** | 46.5 | 65.7 | 80.0 | 15.5 | 29.9 | 46.1 |
| PDCNet+(H) (Truong et al., 2023) | 20.3 | 39.4 | 57.1 | 37.5 | 58.1 | 74.5 | 11.0 | 23.9 | 40.7 |
| Ours | 26.0 | 46.4 | 63.9 | **48.0** | **66.7** | **80.5** | **16.3** | **31.4** | **48.4** |

for DKM and PDCNet+ considering the overall generalization performance and access to the official implementation. Our approach demonstrates the best generalization performance when applying to unseen scenarios, whether indoor (TUM) or outdoor (YFCC) as assessed by the AUC metric at multiple thresholds.

## 5 CONCLUSION AND FUTURE DIRECTIONS

In this paper, we propose a robust sparse and dense matching network termed RSDM incorporating our proposed cascaded GRU refinement module along with an uncertainty estimation module for sparsification. The decoupled training mechanism as well as the increasing diversity of the numerous training data contributes to our superior generalization performance in zero-shot evaluations for both matching and pose estimation. We will further scale up the training data and optimize the performance of downstream geometry estimation.

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

# A APPENDIX

## A.1 SYNTHESIZED TARTANAIR SAMPLED DATASET

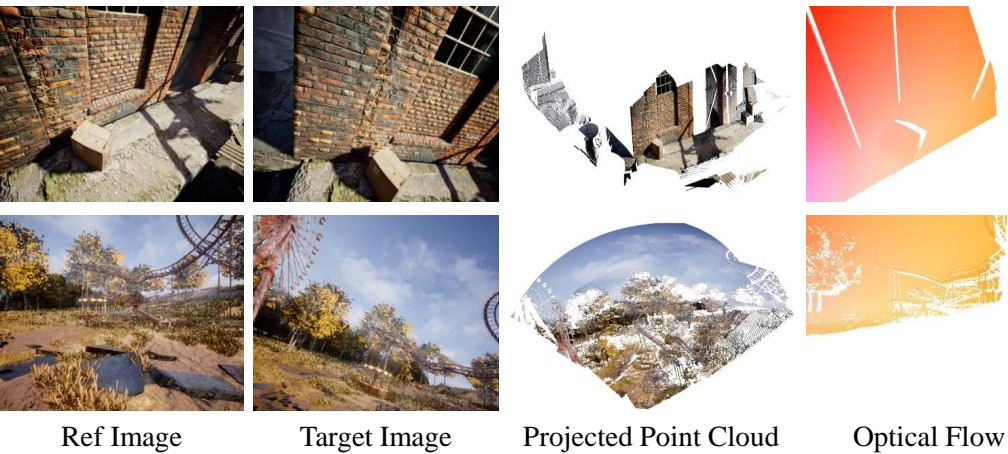

| Ref Image | Target Image | Projected Point Cloud | Optical Flow |

Figure 4: An overview of our generated optical flow sampled from the TartanAir dataset with large intervals to imitate the large viewpoint changes in the real world. The associated changes in brightness as well as the scale of objects can be well preserved.

We organize a set of training data pairs with sparse correspondence annotations based on the TartanAir dataset. The selected reference and target image samples are visualized in FigA.1. The sampling interval is intended to be large for the purpose of significant changes in perspective. We can obtain scale-consistent depth map $\mathbf{D}_{ref}$, intrinsic camera parameter $\mathbf{K}$, camera-to-world camera pose $\mathbf{P}_{ref}$ of the reference image and camera-to-world camera pose $\mathbf{P}_{tar}$ of the target image from original TartanAir dataset. Subsequently, we engage in the projection of pixel coordinates $(u, v)$ from the reference image into 3D point clouds. These points are then further projected onto their corresponding pixels, contingent upon the camera-to-world camera pose $\mathbf{P}_{tar}$ of the target image. The process is depicted as follows.

$$(u', v', 1)^\mathsf{T} = \mathbf{K}\mathbf{P}_{tar}^\mathsf{T}\mathbf{P}_{ref}\mathbf{K}^{-1}\mathbf{D}_{ref}(u, v, 1)^\mathsf{T} \tag{7}$$

where $(u', v')$ is the coordinates of the reference image projected onto the target perspective. After a fundamental coordinate transformation, we can obtain the corresponding grid pairs so that the optical flow in valid regions can be calculated. We subtract the original coordinates $(u, v)$ from the projected coordinates $(u', v')$ to obtain the optical flow $\mathbf{F}$.

$$\mathbf{F} = (u' - u, v' - v) \tag{8}$$

Compared with the synthesized optical flow utilized in previous works (Truong et al., 2020; 2021), our synthesized data with large displacement can preserve the change of brightness and scale introduced by different perspectives.

## A.2 DATA ANALYSIS

As shown in TabA.2, we count the distribution of the Euclidean distance of correspondence on our training and evaluation datasets. The distribution is divided into three intervals [s010, s1040, s40+] indicating the displacement's magnitude falling to 0-10, 10-40, and more than 40 pixels. The real-world datasets like HPatches (Balntas et al., 2017), Megadepth (Li & Snavely, 2018), and Scan-Net (Dai et al., 2017) normally contain a dominant ratio of displacement over 40 pixels compared with the synthesized datasets (Butler et al., 2012; Mayer et al., 2016). It is clear that our synthesized dataset obtains a closer distribution to the real-world datasets which contributes importantly to the robustness of our work.

Table 5: **The distribution of correspondence magnitude.** We count the Euclidean distance on real-world(R) datasets including KITTI, HPatches, Megadepth-1500, and synthesized(S) datasets including FlyingThings3D, TartanAir, and our synthesized dataset. Our generated dataset obtains a greater ratio of large displacement. $R$ is short for real-world while $S$ is short for synthesized.

| Datasets | Type | Distance Distribution | | |
|---|---|---|---|---|
| | | s0-10 | s10-40 | s40+ |
| K (Menze & Geiger, 2015) | R | 30.3 | 36.5 | 33.2 |
| HP (Balntas et al., 2017) | R | 0.9 | 4.5 | 94.6 |
| T (Mayer et al., 2016) | S | 25.2 | 45.7 | 29.1 |
| TA (Wang et al., 2020) | S | 46.2 | 47.2 | 6.6 |
| Si (Butler et al., 2012) | S | 69.0 | 21.3 | 9.7 |
| Sc (Dai et al., 2017) | R | 0.0 | 0.6 | 99.4 |
| M (Li & Snavely, 2018) | R | 1.1 | 10.9 | 88.0 |
| TS(Ours) | S | 0.9 | 9.2 | 89.9 |

### A.3 OPTICAL FLOW ESTIMATION

We conduct experiments for optical flow estimation to explore the performance of dense-based geometry estimation networks and compare our proposed RSDM with other state-of-the-art optical flow methods under the same training pipeline. As represented in TabA.3, generalization results on the Sintel and KITTI datasets are reported with training on the FlyingChairs first following the finetune on the FlyingThings3D datasets. It is clear that DKM struggles in disordered motion estimation and obtains close performance to PWCNet(Sun et al., 2017) as they share a close design of network architecture. The proposed iterative GRU refinement in RAFT(Teed & Deng, 2020) empowers the localization of fast-moving objects which is required in optical flow estimation. Our RSDM achieves competitive results on the Sintel dataset and reaches the lowest outlier percentage on the KITTI dataset. As can be seen in Fig5, the detailed optical flow prediction can be obtained with the help of the employment of fine-grained features.

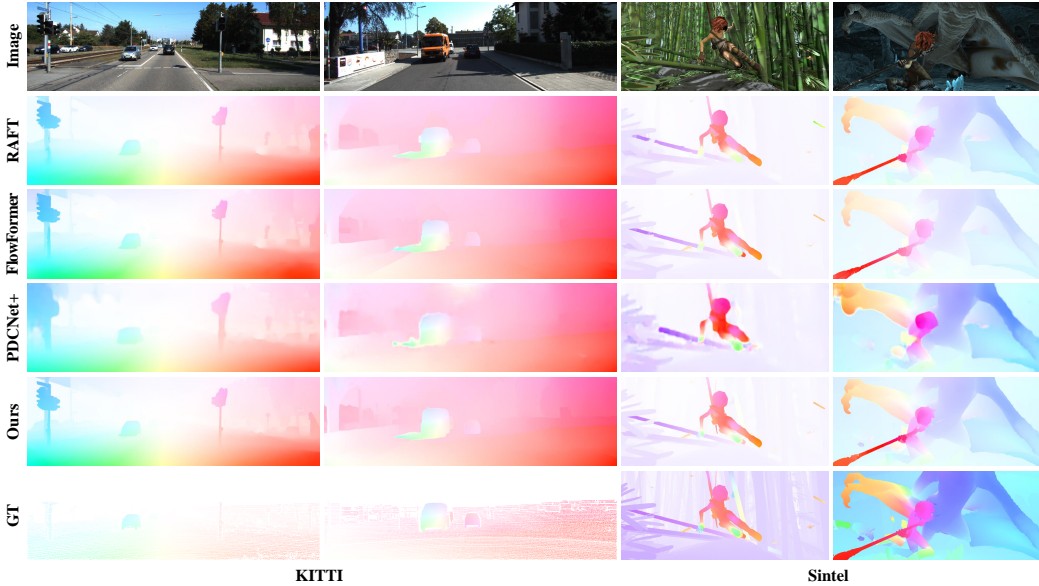

Figure 5: **Qualitative comparison on the KITTI and Sintel datasets.** Our RSDM is capable of estimating more consistent and detailed optical flow compared with others.

Table 6: **Generalization comparison for optical flow estimation.** Following a standard pipeline, we train our RSDM and DKM on FlyingChairs and FlyingThings datasets while testing on Sintel and KITTI datasets. Our RSDM achieves the lowest percentage of matching outliers.

| Method | Sintel-EPE | | KITTI | |
|---|---|---|---|---|
| | Clean | Final | EPE | F1 |
| PWCNet (Sun et al., 2017) | 2.6 | 3.9 | 10.4 | 33.7 |
| DKM (Edstedt et al., 2023) | 2.6 | 4.4 | 13.1 | 37.7 |
| MS-RAFT (Jahedi et al., 2022) | 1.4 | 2.7 | - | - |
| FlowFormer (Huang et al., 2022) | **1.0** | **2.4** | **4.1** | 14.7 |
| GMFlow (Xu et al., 2022b) | 1.1 | 2.5 | 7.8 | 23.4 |
| RAFT (Teed & Deng, 2020) | 1.4 | 2.7 | 5.0 | 17.4 |
| Ours | **1.0** | **2.4** | 4.6 | **13.2** |

A.4    ETH3D EVALUATION

Table 7: **Zero-shot evaluation on the ETH3D dataset.** Both average end-point-error and percentage of correct matches within the 1-pixel threshold are reported at various intervals.

| Method | 5 | 7 | 9 | 11 | 13 | 15 |
|---|---|---|---|---|---|---|
| | AEPE | | | | | |
| RANSAC-Flow (Shen et al., 2020) | 1.9 | 2.1 | 2.3 | 2.4 | 2.6 | 2.8 |
| PDCNet+(H) (Truong et al., 2023) | **1.7** | 2.0 | 2.2 | 2.5 | 2.7 | 3.2 |
| FlowFormer (Huang et al., 2022) | 2.0 | 2.3 | 2.8 | 3.3 | 8.0 | 13.9 |
| FlowFormer++ (Shi et al., 2023) | 2.0 | 2.2 | 2.6 | 3.1 | 4.0 | 8.7 |
| RAFT (Teed & Deng, 2020) | 1.8 | 2.1 | 2.8 | 5.8 | 11.7 | 21.2 |
| DKM (Edstedt et al., 2023) | 1.8 | 1.9 | 2.1 | 2.2 | 2.4 | 2.6 |
| Ours | **1.7** | **1.8** | **2.0** | **2.1** | **2.3** | **2.4** |
| | PCK-1 | | | | | |
| RANSAC-Flow (Shen et al., 2020) | 54.7 | 51.6 | 48.6 | 46.1 | 44.0 | 41.8 |
| PDCNet+(H) (Truong et al., 2023) | 59.9 | 56.8 | 54.1 | 51.6 | 49.6 | 47.3 |
| FlowFormer (Huang et al., 2022) | 55.1 | 50.9 | 47.3 | 44.0 | 40.6 | 37.2 |
| FlowFormer++ (Shi et al., 2023) | 55.2 | 51.1 | 47.5 | 44.1 | 41.2 | 37.7 |
| RAFT (Teed & Deng, 2020) | 56.2 | 52.0 | 48.4 | 44.8 | 41.3 | 37.6 |
| DKM (Edstedt et al., 2023) | 58.9 | 56.0 | 53.4 | 51.1 | 49.2 | 47.1 |
| Ours | **60.8** | **58.2** | **55.9** | **53.8** | **52.0** | **50.3** |

## A.5 VISUALIZATION

We provide more visualizations here.

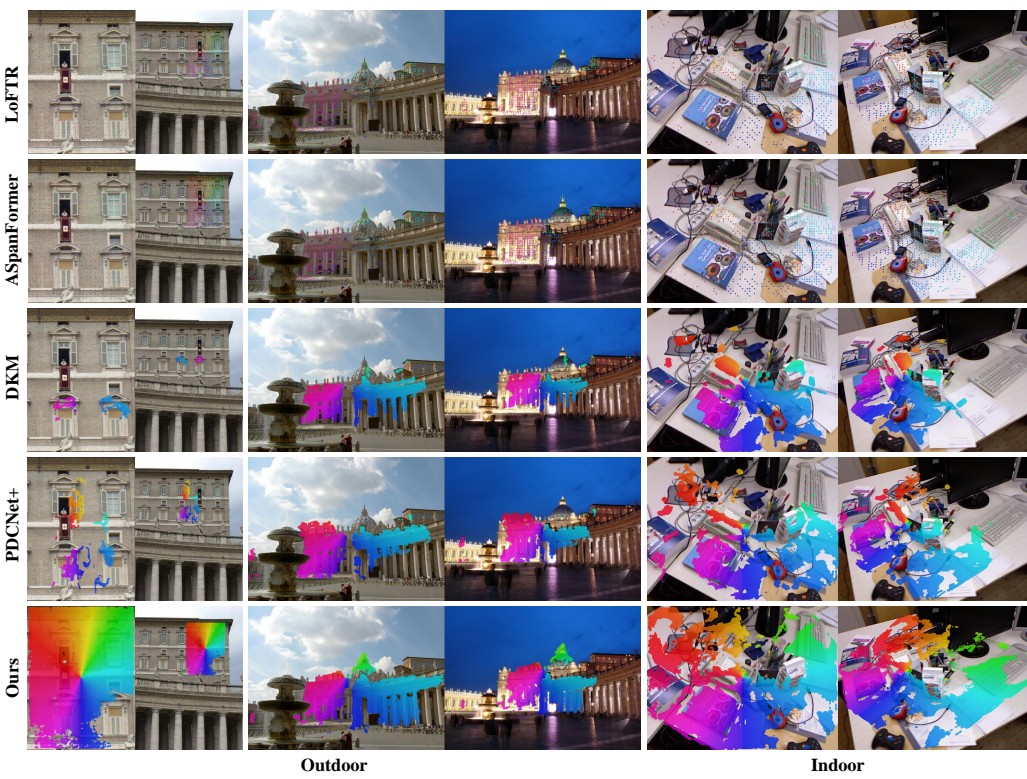

Figure 6: More visualized comparisons between our RSDM and other methods.

