# OpenReview forum: "ROBUST SPARSE AND DENSE MATCHING"
_ICLR.cc/2024/Conference — Submitted to ICLR 2024_

### Official Review · Reviewer_Bj1C · 2023-10-30

**Soundness:** 2 fair
**Presentation:** 3 good
**Contribution:** 2 fair
**Rating:** 5
**Confidence:** 4

**Summary:**

The authors propose a generalized dense matching network, capable of tackling both the optical flow and feature matching tasks simultaneously.  To do this, the authors propose a new model architecture that decouples the uncertainty estimation and investigate how to jointly train on a diverse set of flow and matching datasets.

**Strengths:**

* The authors provide a sound approach to matching, overcoming issues with confidence masks having different interpretations depending on the task by completely decoupling the certainty estimation as a posthoc step. The decoupling is also done in previous work (see PDCNet for example), but they do not detach gradients. What would have perhaps been even more interesting to see would be decoupling of the flow depending on the task. For example, small baseline tasks imply different priors on the types of flow that are likely. There is future potential in this type of approach.

* The paper, except for some minor mistakes, is easy to follow and well written.

* The task is important and a method unifying small and large baseline matching would be of great interest to the community. (See UniMatch)

**Weaknesses:**

* The model architecture is not well motivated. It seems similar to previous optical flow and dense feature matching works. It is not clear why the authors do not start from an established baseline like, e.g., PDCNet or GMFlow.

* The performance on pose estimation is significantly below previous work. The MegaDepth benchmark, which is commonly used, is only shown briefly in the ablation, but no state of the art comparison is provided. The performance is about 5% lower than DKM. On ScanNet the performance is about 3% lower. Also on optical flow the relation to state of the art methods is not documented.

* The ablation on the data is inconclusive. Adding optical flow datasets seem to lower results on pose estimation (Table 1). In the data ablation (Table 2) those results are no longer shown, why? Since those results are not shown, it must be assumed that adding more optical flow datasets further degrade performance.

* Overall message. The manuscript fails in convincing that, with the currently available datasets, unifying wide and small baseline stereo is a good idea. The authors make a good attempt, and their model performs well at both tasks, but worse than the specialized counterparts. Showing that it is possible to do both tasks has been previously shown (GLUnet), so what remains to be shown is that the joint paradigm is superior.

**Questions:**

1. What is the motivation of the architecture choice (see first weakness)?

2. Why does adding optical flow datasets reduce performance (see third weakness)?

---

> ### Author Response · Authors · 2023-11-23
>
> Thanks for your instructive suggestions on our paper. In this work, we aim to improve the generalization performance of the matching problem by collecting large-scale datasets covering optical flow, and geometry estimation datasets. We also propose a generalist model, capable of both optical flow estimation and geometry estimation. Our method achieves the best zero-shot evaluations across all datasets for matching and pose estimation tasks, strongly validating the effectiveness of our proposed method. To mitigate the degeneration of specialized datasets caused by training on data of diverse distributions, we propose a decoupled training strategy by learning matching and uncertainty estimation in a two-stage manner. Overall, we have taken into full consideration both the specificity and generality. The best zero-shot generalization performance compared with prior generalist and specialized models as well as the competitive performance compared with specialized methods can strongly demonstrate the contributions of our work. We also report the performance of optical flow following the standard setting as specialized models. The comparison is illustrated in Tab.6. We provide the updated results by adopting the tile mechanism proposed in FlowFormer. The results are shown below:
>
> | Methods       | Sintel-C &darr; | Sintel-F &darr;  | KITTI EPE &darr; | KITTI F1 &darr; |
> | ------------- | --------------- | ---------- | -------- | ---------- |
> |PWCNet| 2.6 | 3.9 | 10.4 | 33.7 |
> |DKM| 2.6 | 4.4 | 13.1 | 37.7 |
> |MS-RAFT| 1.4 | 2.7 | - | - |
> |FlowFormer| **1.0** | **2.4** | 4.1 | 14.7 |
> |GMFlow| 1.1 | 2.5 | 7.8 | 23.4 |
> |RAFT| 1.4 | 2.7 | 5.0 | 17.4 |
> |Ours| **1.0** | **2.4** | **3.9** | **12.6** |
>
> This comparison strongly validates that if training on specific datasets under the same setting as specialized models, our method can also obtain superior performance. We will provide detailed explanations to your problems below:
>
> ## Network architecture:
> Thanks for pointing out this problem. We provide a detailed explanation for this problem in our first official comment. The primary reasons are twofold. First of all, the iterative optimization mechanism provides an explicit and efficient approach for exploiting geometric similarity, which can be employed in geometry estimation methods for further improvement. Moreover, conducting matching at a higher resolution can empirically introduce additional improvement for downstream pose estimation as in DKM, PATS, etc. However, popular optical flow estimation works like RAFT, FlowFormer, and GMFlow end up with the refinement at a relatively low resolution(1/4 at most). To mitigate this problem, we elaborately construct the model architecture to perform refinement at 1/2 resolution. In future work, we will conduct more comprehensive experiments and utilize your mentioned models(GMFlow, PDCNet+) for generalization evaluations with our training data.
>
> ## Degeneration with optical flow
> Currently, widely used datasets for training optical flow, such as FlyingThings3D, FlyingChairs, Sintel, and TartanAir, are synthesized datasets with relatively small displacements. In contrast, real-world datasets for geometry estimation often exhibit substantial changes in viewpoint. The integration of a diverse set of datasets with varying distributions, as opposed to training on specific datasets tailored for specialized tasks, may introduce inevitable interference, which accounts for the degeneration. The ablation studies in Table 1 are primarily conducted to reveal this phenomenon and demonstrate the effectiveness of our proposed decoupled training strategy in mitigating this decline. A more straightforward table for downstream pose estimation is presented below:
>
> |Methods|Datasets|Megadepth AUC@5|YFCC AUC@5|
> |---|----|----|----|
> |Joint|M|55.3|48.2|
> |Joint|M+TS|51.9|47.8|
> |Decoupled|M|55.7|48.5|
> |Decoupled|M+TS|54.0|48.2|
>
> Leveraging our decoupled training strategy efficiently enhances the performance when the optical flow dataset is used. In Tab.2, we utilize our decoupled training strategy in default to explore the effectiveness of scaling up the diversity of training for generalization performance. The uncertainty estimation module is not involved at this matching stage so only generalization evaluations on matching tasks are reported.

---

> > ### Comment · Reviewer_Bj1C · 2023-11-23
> >
> > I thank the authors for their rebuttal and will check my assessment after reading the responses to all reviews.

---

> ### Author Response · Authors · 2023-11-23
>
> We greatly appreciate your immediate correspondence. We sincerely hope that in your reassessment, you can kindly take our primary contributions to the generalized matching task for both matching and pose estimation as well as our explorations to alleviate the degeneration when training on the mixed data of diverse distributions into consideration. Thanks for your instructive and academic suggestions on our work again!

---

### Official Review · Reviewer_Sn8S · 2023-10-31

**Soundness:** 3 good
**Presentation:** 3 good
**Contribution:** 3 good
**Rating:** 5
**Confidence:** 3

**Summary:**

This paper propose a robust sparse and dense matching network termed RSDM which can generalize well to unseen scenarios with our proposed cascaded GRU refinement for dense correspondence estimation and an uncertainty estimation module for sparsification.  The authors explore the effectiveness of scaling up the training data by mixing up multiple datasets. A comprehensive analysis is conducted to explore a more reasonable training strategy for enhanced robustness. The RSDM achieves state-of-the-art generalization performance in zero-shot evaluations for both matching and geometry estimation across multiple datasets, outperforming previous generalist and specialized models by an obvious margin

**Strengths:**

This paper propose a robust sparse and dense matching network termed RSDM incorporating the proposed cascaded GRU refinement module along with an uncertainty estimation module for sparsification. The decoupled training mechanism as well as the increasing diversity of the numerous training data contributes to the superior generalization performance in zero-shot evaluations for both matching and pose estimation.
The strengths are as follows:
1. The proposed RSDM can deal with both  sparse and dense matching task
2. The proposed method mix up various dense and sparse matching datasets which significantly improves the training diversity.
3. Superior performance is achieved for zero-shot matching as well as downstream geometry estimation across multiple datasets, outperforming the previous methods by a large margin

**Weaknesses:**

The weakness are as follows:
1. The proposed model use high-weight parameters, swin-transformer, RAFT. It doesn't present the comparison with other methods.
2. The "Warping Error Map" is not detailed in paper, but it's important
3. How to use "Uncertainty Map" in ransac filter, it should be given in detail.
4. In the experiments, the proposed method achieves good performance on zero-shot matching evaluations. but for Downstream pose estimation, it works not very well. Compared with DKM, its result is not very good. but the authors has no explanation.
5. There is no model size and runtime cost comparison with other methods.

**Questions:**

No questions

---

> ### Author Response · Authors · 2023-11-23
> **Correspondence to reviewer2**
>
> Thanks for pointing out the weakness of our work and we apologize for the unclear writing in this component. We will provide more comprehensive experiments in our revised supplementary materials.
>
> ## Comparison with other methods:
> In our work, we utilize the shifted-window strategy to perform self-attention and cross-attention for feature enhancement. The compared methods like GMFlow and FlowFormer also adopt the high-weight parameters. It's worth mentioning that FlowFlower utilizes the Twins-L model as its backbone for feature extraction.
>
> ## Warping error computation
> As briefly introduced in Sec.3.2, we warp the target frame to the reference frame according to the estimated optical flow with bilinear sampling. Given the input reference and target images $I_r, I_t,$ as well as the estimated optical flow $f$ at the grid $x$, the warped reference image can be computed as:
> $$
> I_{warped}(x) = I_r(x+f(x)),
> $$
> and the warped RGB error can be computed as:
> $$
> E_{RGB} = |I_r - I_{warped}|
> $$
> We also compute the feature error $E_{feat}$ in the same way with reference and target features $F_r, F_t$:
> $$
> E_{feat}(x) = |F_r(x) - F_t(x+f(x))|
> $$
>
> The final warping error is defined as the concatenation of both warped RGB error and warped feature error. We will add detailed introduction about the implementation of warping error in our revised paper.
>
> ## Uncertainty map for ransac filter
> The uncertainty map serves as a manifestation of the effectiveness of sparse correspondence matching. A higher degree of effectiveness in the sparse matching outcomes for a specific point is associated with an increased magnitude of its corresponding uncertainty. The cumulative uncertainties of all points within an image collectively contribute to the formation of its corresponding uncertainty map. We conduct a point selection procedure involving the uncertainty map and its corresponding matching outcomes, subsequently directing the selected points into a RANSAC filter. Our point selection process comprises two sequential steps. The initial step involves the selection of points using uncertainty as a screening criterion. The second step, based on the distribution among these selected points, constitutes a secondary selection aimed at ensuring diversity in the chosen points. In the specific context of the secondary point screening step, it entails the initial computation of Euclidean distances among the points selected during the initial screening. Euclidean distances are subsequently employed as a reference to establish weights for the secondary point selection, where greater distances correspond to augmented weights. As a preventive measure against the introduction of noise data, weights for points with distances exceeding a predefined threshold are reset to the minimum value of 1e-7.
>
> ## Comparison with other optical flow methods:
> Sorry for the shortage of this comparison with other optical flow methods in the main text which is placed in the supplementary materials as can be found in Tab.6. In our later research, we find that additional improvement can be obtained by adopting the tile technique mentioned in FlowFormer for alleviating the influence of position embedding at different resolutions. We provide the updated results in the following table:
>
> | Methods       | Sintel-C &darr; | Sintel-F &darr;  | KITTI EPE &darr; | KITTI F1 &darr; |
> | ------------- | --------------- | ---------- | -------- | ---------- |
> |PWCNet| 2.6 | 3.9 | 10.4 | 33.7 |
> |DKM| 2.6 | 4.4 | 13.1 | 37.7 |
> |MS-RAFT| 1.4 | 2.7 | - | - |
> |FlowFormer| **1.0** | **2.4** | 4.1 | 14.7 |
> |GMFlow| 1.1 | 2.5 | 7.8 | 23.4 |
> |RAFT| 1.4 | 2.7 | 5.0 | 17.4 |
> |Ours| **1.0** | **2.4** | **3.9** | **12.6** |
>
> This comparison strongly validates that when training on specific datasets for specialized tasks, our model still obtains outstanding performance, achieving the best results on both Sintel and KITTI datasets. We will provide comprehensive comparisons with other specialized models under the same setting of specific tasks to validate the effectiveness of the proposed method. Moreover, we will place the aforementioned comparison on optical flow estimation in our main text in the revised version.
>
> ## Computation cost and inference speed
> Thanks for pointing out the absence of this experiment. We select state-of-the-art methods and make comparisons with them in terms of computation costs and inference speed. The result is shown below:
>
> |Methods|Computation Costs(GFLOPs)|Inference Speed(s)|
> |---|----|----|
> |FlowFormer|695.0|0.23|
> |GMFlow|202.8|0.06|
> |RAFT|505.8|0.15|
> |Ours|940.7|0.19|
>
> We will explore to construct a more efficient model in our future work.

---

### Official Review · Reviewer_gGcr · 2023-11-01

**Soundness:** 2 fair
**Presentation:** 3 good
**Contribution:** 2 fair
**Rating:** 6
**Confidence:** 4

**Summary:**

This work proposed the robust network suitable for both sparse and dense matching tasks called RSDM. In this work, simlarity matrix/cost volume of three scales are generated with feature level refinement and GRU based correlation volume level refinement. Context level information is also used to guide the GRU refinemnet block for the first scale.  For sparsification, warp error based on predicted dense matching results are used to estimate the uncertainty while balanced sampling strategy are use. This work also generate a dataset based on TartanAir with optical flows generated. Experiments are hold based on several banchmarks outperforming the previous methods by a large margin however several experimental results have to be provided.

**Strengths:**

1) Using flow to achieve cross-scale transfer of matching relationships is an efficient solution.
2) The ability to scale to diverse downstream tasks makes this approach attractive.

**Weaknesses:**

1. The problem statement of  "robust for sparse and dense matching":

   What are the main differences between the RSDM and the methods only fit for sparse or dense matching task? The RSDM seems designed based on original dense metching pipelines such as GMFlow with uncertainty estimation(from DKM) for sparsifiy the dense matching result. Can this setting be used in other dense matching works to make it suitable for sparse matching tasks?

2. The effectiveness of multi-scale design:

   The method used the FPN and  generate simlarity matrix in three scales. However, in the following three GRU Refinement Blocks only one  matrix seemes to be used. How about the matrixes in other two scales. Besides, further ablations on the multi-scale design should be provided.

3. The design of dataset:

   The proposed dataset seems like a subset of TartanAir dataset with a improved optical flow rendering method. What is the main problem solved by building this data set? What are the advantages over previous datasets besides better optical flow supervision? More experimental results based on this dataset need to be given.

4. Several results in ablation study is not clear:

   The data in the experimental table cannot clearly reflect the effectiveness of each module. For example, Table 1, what is the setting of RSDM? Is it the last row?

**Questions:**

See the Weakness part

---

> ### Author Response · Authors · 2023-11-23
> **Correspondence to Reviewer1**
>
> Thanks for pointing out the problems of our work. Here are our explanations.
> ## Model architecture
> As mentioned in the introduction, the model design is not our primary contribution in this work. We think any dense matching framework coupled with an uncertainty estimation module can serve as a generalist model. However, currently available models have their limitations. The iterative optimization mechanism provides an explicit and efficient approach for exploiting geometric similarity, which can be employed for geometry estimation. Emperiaclly, conducting matching at a higher resolution can introduce additional improvement for downstream pose estimation as in DKM, PATS, etc. However, popular optical flow estimation works like RAFT, FlowFormer, and GMFlow end up with the refinement at a relatively low resolution(1/4 at most), which is the main reason we elaborately construct the model architecture to perform refinement at 1/2 resolution.
>
> ## Advantage of our organized dataset:
> Thanks for pointing out this question. The re-organized of the TartanAir dataset is an important exploration in our work. Notably, the magnitude of displacement varies substantially across different datasets and scenarios. We list the displacement distribution of different datasets as well as our organized data below which can also be found in Tab.5 in our paper. It's worth noticing that our generated optical flow dataset obtains a different distribution compared to the original TartanAir dataset.
>
> |Datasets|Task|s0-10|s10-40|s40+|
> |----|----|-----|-----|--------|
> |KITTI|Optical Flow|30.3|36.5|33.2|
> |HPatches|Geometry Estimation|0.9|4.5|94.6|
> |FlyingThings3D|Optical Flow|25.2|45.7|29.1|
> |TartanAir|Optical Flow|46.2|47.2|6.6|
> |Sintel|Optical Flow|69.0|21.3|9.7|
> |ScanNet|Geometry Estimation|0.0|0.6|99.4|
> |Megadepth|Geometry Estimation|1.1|10.9|88.0|
> |Ours|Optical Flow|0.9|9.2|89.9|
>
> We re-organize the dataset by increasing the sampling interval to obtain a closer distribution of other real-world datasets. The computation of optical flow ground truth is illustrated in our appendix A.1. The effectiveness of our proposed dataset is obvious as shown in Tab.1 and Tab.2. We simplify Tab.2 as below to show the effectiveness of our organized TartanAir sampled(TS) dataset.
>
> | With TS dataset       | HPatches PCK-1 &uarr; | KITTI AEPE&darr;  | KITTI F1 &darr; | ETH3D AEPE &darr; | ETH3D PCK-1 &uarr; |
> | ------------- | --------------- | ---------- | -------- | ---------- | ----------- |
> | &#10060;   | 44.3            | 4.1        | 10.8     | **2.0**    | 55.9        |
> | &#10004; | **46.3**        | **3.5**    | **9.6**  | **2.0**    | **56.3**     |
>
> The adoption of our organized TartanAir dataset efficiently improves the generalization performance across multiple datasets, demonstrating the advantages of our organized dataset.
>
> ## Ablations on multi-scale design
> We are sorry for the confusion caused by the absence of this experiment. We provide the following ablation studies by inferring at previous scales for a quick validation of our multiscale design. The results are shown below and scale 3 corresponds to the refinement of 1/2 resolution utilized in our final model while scale 1 indicates the refinement of 1/8 resolution:
>
> | Output Scale       | HPatches PCK-1 &uarr; | KITTI AEPE&darr;  | KITTI F1 &darr; | ETH3D AEPE &darr; | ETH3D PCK-1 &uarr; |
> | ------------- | --------------- | ---------- | -------- | ---------- | ----------- |
> | 1   | 40.4 | 4.9 | 15.1 | 11.3 | 40.4 |
> | 2 | 46.5 | 4.5 | 11.1 | 10.1 | 46.5 |
> | 3 | **47.3** | **4.4** | **10.9** | **9.9** | **47.3** |
>
> Furthermore, we also utilize the matching at the previous scale for pose estimation, and the performance is shown below:
> |Output Scale| ScanNet AUC@5 | YFCC@5 | TUM@5 |
> |-------|-------|-------|-------|
> |1| 22.9 |39.4|15.0|
> |2| 25.3 |47.0|16.0|
> |3| **26.0** |**48.3**|**16.3**|
>
> The above-listed results across matching and pose estimation tasks validate the effectiveness of the multi-scale design and the necessity of applying refinement at a higher resolution. We will place this comparison in the supplementary materials and conduct more comprehensive ablation experiments in our revised paper.
>
> ## Ablation Study Setting:
> We apologize for the ambiguity caused by our experiment settings and paper writing. In Tab.1, we illustrate the effectiveness of the decoupled training strategy by learning matching and uncertainty in a two-stage manner. In our final experiment, we adopt this decoupled training strategy. At the matching stage, we first train the dense matching model on the mixture of datasets as shown in Tab.2 to improve the generalization performance, and the effectiveness is validated. In the second stage, we freeze the parameters of the matching model and train the uncertainty estimation model based on the warping error according to the matching model. At this stage, only Megadepth and ScanNet are utilized. The detailed training settings are mentioned in Sec.4.1.

---

> > ### Comment · Reviewer_gGcr · 2023-12-03
> >
> > After thoroughly reviewing the authors' rebuttal and considering the feedback from other reviewers, I appreciate the detailed responses provided to address the concerns raised. Based on these considerations, I maintain my initial rating for this submission.

---

### Author Response · Authors · 2023-11-23
**Explanations to common concerns**

Dear reviewers, we appreciate your instructive suggestions on our work, and we apologize for the confusion caused by the ambiguities in our writing. We will modify the unclear components in our revised paper and provide necessary experiments for more comprehensive comparisons in our supplementary materials.

For the commonly concerned problem of our degeneration compared with specialized models on downstream pose estimation, we want to make the following explanations and re-emphasize the motivation and contributions of our work.

## Degeneration on downstream tasks.

While specialized models demonstrate remarkable accuracy in specific tasks, their generalization capacity is constrained, which is a primary challenge we aim to tackle in this work. An efficient approach is to scale up the diversity of training data. Compared with prior generalized models like PDCNet+ and GLUNet, we collect more training data from optical flow and geometry estimation tasks. The improvement in genereralization performance is strongly validated in our experiments on both matching and pose estimation tasks which can be found in Tab.3 and Tab.4. Our proposed method obtains the best performance in zero-shot evaluations for both matching and pose estimation, surpassing previous generalized models by a significant margin.

However, the introduction of additional data with diverse distributions and scenarios can introduce interference when compared to training exclusively on specific data tailored for specialized tasks. This interference accounts for the degradation of our method in downstream pose estimation compared to models trained on specific datasets designed for such tasks. This domain conflict has existed for a long time and remains a great challenge. To alleviate this degeneration, we propose the decoupled training strategy and the effectiveness of our decoupled training strategy can be found in Tab.2. When comparing with training solely on the Megadepth dataset, the joint learning of matching and uncertainty estimation on Megadepth and our organized TartanAir datasets significantly decreases the AUC@5 metric from 55.3 to 51.9 and 48.2 to 47.8 on Megadepth and YFCC datasets, respectively. Compared with the downstream pose estimation, the degradation on the matching is relatively small. We attribute this to the negative impact on uncertainty estimation induced by the incorporation of our organized dataset. To mitigate this, we decouple the learning of matching and uncertainty estimation into two stages. We introduce additional data of diverse distributions to train the matching model. At the uncertainty estimation stage, only geometry estimation datasets are adopted. We simplify Tab.2 and provide a more intuitive comparison of pose estimation as below:

|Methods|Datasets|Megadepth AUC@5|YFCC AUC@5|
|---|----|----|----|
|Joint|M|55.3|48.2|
|Joint|M+TS|51.9|47.8|
|Decoupled|M|55.7|48.5|
|Decoupled|M+TS|54.0|48.2|

Analysis of the aforementioned table reveals that our decoupled training strategy effectively mitigates the degradation introduced by the incorporation of additional data. We will further explore more effective approaches to minimize discrepancies when compared to specialized models while preserving the generalization performance in our future work.

It's worth emphasizing our method obtains the best generalization performance of pose estimation as well:

|Methods|YFCC AUC@5|YFCC AUC@10|YFCC AUC@20|TUM AUC@5|TUM AUC@10|TUM AUC@20|
|---|----|----|----|----|----|----|
|LoFTR| 42.4 | 62.5 | 77.3 | - | - | - |
|MatchFormer| 53.3 | 69.7 | 81.8 | - | - | - |
|ASpanFormer| 44.5 | 63.8 | 78.4 | - | - | - |
|PATS| 47.0 | 65.3 | 79.2 | - | - | - |
|DKM| 46.5 | 65.7 | 80.0 | 15.5 | 29.9 | 46.1 |
|PDCNet+| 37.5 | 58.1 | 74.5 | 11.0 | 23.9 | 40.7 |
|Ours| **48.0** | **66.7** | **80.5** | **16.3** | **31.4** | **48.4** |


## Model architecture
As mentioned in the introduction, the model design is not our primary contribution in this work. We think any dense matching framework coupled with an uncertainty estimation module can serve as a generalist model. However, currently available models have their limitations. The iterative optimization mechanism provides an explicit and efficient approach for exploiting geometric similarity, which can be employed for geometry estimation. Emperiaclly, conducting matching at a higher resolution can introduce additional improvement for downstream pose estimation as in DKM, PATS, etc. However, popular optical flow estimation works like RAFT, FlowFormer, and GMFlow end up with the refinement at a relatively low resolution(1/4 at most), which is the main reason we elaborately construct the model architecture to perform refinement at 1/2 resolution.

---

### Meta-Review · Area_Chair_QY1n · 2023-12-06

**Metareview:**

There is a lot to like about this paper. It's long overdue to merge dense optical flow and sparse feature matching into one generalist framework (although it was somewhat done before), and it would be a slam dunk if the proposed RSDM indeed achieved superior performance on both tasks. The proposed method, focusing on an elaborately decoupled training strategy, certainly has the merit of carefully dealing with the discrepancy between wide and narrow-baseline matching datasets to avoid interference (but the experiments showed such interference persists after adding the flow datasets).

The AC wishes the authors could have really focused on answering reviewers’ questions. For instance, R# Bj1C raised very specific concerns that unifying both wide and small baseline stereo has been shown by GLUnet, so what would be expected is really a joint paradigm is superior. However, the authors didn’t address these critical points during the rebuttal and hence the reviewers were not convinced. Likewise, the authors also didn’t address the concerns raised by R# Sn8S on the degradation of downstream geometry estimation across multiple datasets (only mentioned that conducting matching at a higher resolution can introduce additional improvement in addressing other reviewers).

While the AC really likes the goal of the paper, it is not ready for publication in its current stage. There are probably more important recipes to be discovered for this paper to really shine, and the AC cannot wait for that day!

**Justification For Why Not Higher Score:**

The proposed method only shined on zero-shot, but not each specific task. As pointed out by R# Bj1C, such joint task was already proposed in GLUnet, and therefore the paper is expected to report a "generalist model" that outperforms the specialist (at least on par).

**Justification For Why Not Lower Score:**

N/A

---

### Decision · Program_Chairs · 2024-01-16

Reject